# The Effects of Sternocleidomastoid Muscle Taping on Postural Control in Healthy Young Adults: A Pilot Crossover Study

**DOI:** 10.3390/healthcare10050946

**Published:** 2022-05-20

**Authors:** Alex Martino Cinnera, Alessandro Antonio Princi, Enza Leone, Serena Marrano, Alessandra Pucello, Stefano Paolucci, Marco Iosa, Giovanni Morone

**Affiliations:** 1Scientific Institute for Research, Hospitalization and Health Care (IRCCS) Santa Lucia Foundation, Via Ardeatina, 00179 Rome, Italy; a.princi@hsantalucia.it (A.A.P.); sere.marrano@gmail.com (S.M.); pucelloalessandra@gmail.com (A.P.); s.paolucci@hsantalucia.it (S.P.); 2School of Allied Health Professions, Keele University, Staffordshire ST5 5BG, UK; e.leone@keele.ac.uk; 3Centre for Biomechanics and Rehabilitation Technologies, Staffordshire University, Stoke on Trent ST4 2DE, UK; 4Department of Psychology, Sapienza University of Rome, 00185 Rome, Italy; m.iosa@hsantalucia.it; 5Department of Life, Health and Environmental Sciences, University of L’Aquila, 67100 L’Aquila, Italy; giovanni.morone@univaq.it

**Keywords:** posture, postural control, tape, bandage, kinesio-tape, equipment, supplies

## Abstract

Background: Postural control is a complex ability, also controlled by the somatosensory connection of the neck muscles with the vestibular nuclei. This circuit seems to be interested in maintaining head stabilization during movements. The sternocleidomastoid (SCM) muscle is the dominant source of the vestibular afferents as confirmed by neurophysiological acquisition. The aim of this study is to evaluate whether the application of kinesio-tape on the SCM muscle can induce a perturbation of the standing postural control by altering the somatosensory system of the neck muscles. Methods: Thirteen healthy participants (age: 24.46 ± 3.04 yrs; 9 female) were enrolled, and the four kinesio-tape (KT) conditions were performed in a random order: without KT application (Ctrl); right KT application (R-SCM); left KT application (L-SCM); and bilateral KT application (B-SCM). All conditions were performed three times with open eyes and closed eyes. Results: There was a significant increase in the length of the centre of pressure (CoP), in the maximal oscillation, and in the anteroposterior root mean square between the three tape application conditions with respect to the Ctrl condition with open eyes. The same parameters were statistically different when the participants were blindfolded in the B-SCM condition with respect to the Ctrl condition. A statistical decrease in the difference in weight distribution between the two feet was observed in the B-SCM group with respect to the Ctrl group in both open and closed eyes conditions. Conclusions: Our results suggest that KT on the SCM muscles may involve some space-time parameters of postural control. Bilateral KT improved the weight distribution between the feet but showed a parallel increase in anteroposterior oscillations and in the length of the CoP with respect to the Ctrl condition. The perturbation seems to be greater in the somatosensory system when it is working coupled with visual afferences during an upright position.

## 1. Introduction

Postural control is defined as “the act of maintaining, achieving, or restoring a state of balance during any posture or activity” [1] (p. 402), [2]. Alterations of the balance control may be associated with ear symptoms, visual symptoms (i.e., diplopia and oscillopsia), vertigo, and dizziness. Vestibular disorders [3], chronic neurological diseases [4], and musculoskeletal impairments [5] might be frequently the causes of these symptoms. These symptoms should not be continuous, but they might occur during head or body movements for example during activities of daily living and might become chronic [6]. Fundamentally, standing upright requires individuals to distribute their unstable whole-body load within a small base of support. Small perturbations of the upright body are detected by visual, vestibular, and somatosensory receptors [7,8], which encode these movements through their own coordinate system through specific dynamics. The balance controller filters, processes, and integrates sensory cues of body motion to produce an error signal between the predicted and actual sensory consequences of balance-related movements. In response to any mismatch between the predicted and actual consequences, compensatory motor commands are generated to maintain the upright standing position [9]. Therefore, postural control can be viewed as a closed-loop feedback control system with the integration of vestibular, visual, and proprioceptive information for spatial orientation [10], which are regulated by the sensory integration process system [11]. The upright standing position is unstable, as any external perturbation to the upright orientation produces forces that accelerate the body. Deviations from the desired orientations are detected by the sensory systems (primarily somatosensory/proprioceptive, visual, and vestibular systems), which, in response, generate appropriate joint torques to preserve stability [11]. A deficit in one or more of these receptor systems or in the sensory integration can cause alterations of stability in upright postural control. Part of this postural system is controlled by the somatosensory connection of the neck muscles with the vestibular nuclei and cerebellum. Specifically, this circuit is interested in maintaining head stabilization during movements. The head stabilization reflex, which manifests in the neck muscles, is triggered by sudden head position changes and is responsible for returning the head to its previous position. The head stabilization reflex consists of afferent fibres coming from the cervical muscle spindles, vestibular structures, and the accessory nerve, and the efferent fibres from the accessory nerve [12]. This closed-loop circuit (vestibulocollic reflexes) can be explained by vestibular evoked myogenic potentials (VEMPs); these otolith-dependent reflexes are produced by stimulating the ears with air-conducted sound or skull vibration and are recorded from surface electrodes placed over the neck (cervical VEMPs) and eye muscles (ocular VEMPs) [13]. The sternocleidomastoid muscle (SCM) is the predominant source of the cervical VEMPs [14]. In fact, this relationship has been confirmed via changes in motor units of the SCM that are evoked by the same stimuli (electric or acoustic) capable of evoking vestibulocollic reflexes [15]. Specifically, nervous projections from the vestibular system to the SCM are predominantly from the saccular macula [16]. As a result of postural control, the SCM muscles are activated following a perturbation [17]. In recent years, clinicians have often used kinesiology tape (kinesio-tape) as a treatment strategy to support the neck fascia, muscles, and joints to reduce cervical pain [18,19]. By stimulating the mechanoreceptors in the skin and, consequently, triggering reflex action, kinesio-tape can also cause changes in the muscle tone and enhance proprioception [20]. Available evidence has shown that the application of kinesio-tape in healthy individuals positively affected their postural control [21]. However, in most of the studies, the effects of kinesio-tape on postural control and dynamic balance were assessed by applying the kinesio-tape to lower extremity structures, especially ankles [22]. Therefore, the effects that the kinesio-tape applied to the SCM may have on the postural control in healthy individuals remain unexplored.

The aim of this study is to test whether the application of kinesio-tape on the SCM muscles can induce a perturbation of the standing postural control in healthy young adults by altering the somatosensory system of the neck muscles.

## 2. Materials and Methods

### 2.1. Study Design

A study having a within-subjects design was performed on healthy individuals. The study was conducted in accordance with the Declaration of Helsinki and was approved by the independent committee of Santa Lucia Foundation Hospital. All participants provided written informed consent prior to enrolling in the study. A convenience sampling of healthy individuals without history of neurological diseases or lower limb injuries that could interfere with the study was included. Thirteen participants (9 female; 24.46 ± 3.04 years of age) were tested in four conditions in a random order: (1) without kinesio-tape (Ctrl); (2) with kinesio-tape on the right SCM muscle (R-SCM); (3) with kinesio-tape on the left SCM muscle (L-SCM); and (4) with kinesio-tape on both SCM muscles (B-SCM) (complete demographics information of the sample are available in Table 1).

### 2.2. Tape Application

A certified kinesio-tape practitioner administered all taping procedures. Kinesio-tape (Kinesiology Human Tape, Prosomed^®^, Bolzano, Italy) with a Y-shape application was applied along the course of the muscle belly, from the mastoid process of the temporal bone without strain. The patients turned their head away to the side in which the tape was to be applied and bent their necks laterally. The tails of the tape were attached to the manubrium of the sternum (tail 1) and ^1^⁄₃ medial of the clavicle (tail 2) [23] (Figure 1).

### 2.3. Stabilometric Assessment

Postural control was assessed using a 320-cm by 75-cm (length × width) static force platform (Platform BPM 120, Physical Support Italia, Rome, Italy). The signals were amplified and acquired using the Physical gait Software Vv. 2.66, Physical Support Italia, Rome, Italy. Spatiotemporal parameters of static postural control were assessed with participants standing barefoot on a force platform. Participants were asked to maintain a relaxed upright position with their arms by their sides. Feet were placed with the forefoot turned out at 30 degrees and the heels at a comfortable distance. The measurements were conducted under the four experimental conditions (Ctrl, R-SCM, L-SCM, and B-SCM). Each condition was assessed, in a random order, three times with open eyes (OE), while facing a target placed 1.5 m away from them, and three times with closed eyes (CE). The testing time was 51.2 s based on the indications of the platform manufacturer and in agreement with previous studies [24,25,26]. The length of the centre of pressure (CoP) trajectory (mm) was measured to obtain the overall CoP length. The body weight distribution between the two sides was reported in terms of percentage. All participants with visual impairment performed the test wearing personal glasses or contact lenses.

### 2.4. Statistical Analysis

Because of the small sample size and the non-normality of the data, which were tested using Shapiro-Wilks’s test and a histogram, one-way repeated measures of analysis of variance with the nonparametric Friedman statistical test were used to detect differences between the four conditions (mean of the three repetitions). All results with a *p* value less than 0.05 were investigated by pairwise comparison with the Bonferroni correction. All pairwise results with *p* < 0.0125 (0.05/4) were considered statistically valid. The effect size of the significant results was calculated with the Cohen method (Cohen’s d = Mean 1 − Mean 2/√ (SD1^2^ + SD2^2^)/2) and interpreted as small (d  = 0.2), medium (d  = 0.5), and large (d ≥ 0.8) [27]. Statistical analysis was conducted using Statistical Package for Social Science (SPSS) software (IBM SPSS Statistics Version 25).

## 3. Results

There was a significant statistical difference in the length of the CoP when the participants performed the test with open eyes, χ^2^(3) = 18.692, *p* < 0.0001. Post hoc analysis showed a statistically significant increase between the three tape application conditions with respect to the control condition (L-SCM vs. Ctrl, *p* = 0.05; R-SCM vs. Ctrl, *p* = 0.003; and B-SCM vs. Ctrl, *p* < 0.0001). There was also a statistically significant increase in the length of the CoP when the participants were blindfolded, χ^2^(3) = 8674, *p* < 0.03. Post hoc analysis showed a statistically significant difference only between the condition in which tape was applied bilaterally with respect to the control condition (*p* = 0.04). A significant statistical difference was observed in the maximal oscillation when the participants performed the test with open eyes, χ^2^(3) = 13,093, *p* < 0.004. Post hoc analysis showed a significant statistical difference between the three tape application conditions and the control condition (L-SCM vs. Ctrl, *p* = 0.01; R-SCM vs. Ctrl, *p* = 0.03; and B-SCM vs. Ctrl, *p* < 0.01). These oscillations seem to be confirmed in an increase in the root mean square (RMS) in the anteroposterior axis (Y), in both eyes’ conditions, especially in the bilateral tape application. Interestingly, the difference in weight distribution between the two feet showed a statistical decrease when the participants performed the test with B-SCM tape with respect to the control condition with open eyes (B-SCM = 5.08%; Ctrl = 10.23%) and closed eyes (B-SCM = 4.92%; Ctrl = 8.62%). No other interesting differences were found between the four conditions in the other spatiotemporal parameters examined with computerised posturography (the complete results are presented in Table 2). Effect size analysis showed a general medium to large effect in all significant results. Specifically, a large effect was found in the length of the CoP and in the anteroposterior oscillation (RMS in the Y axis) with respect to the control condition and KT applications, both with open eyes and closed eyes (the complete effect sizes of significant results are reported in Table 3).

## 4. Discussion

Whether we are engaged in a dynamic or static balance task, the neck muscles maintain the upright orientation of the head over the body. For this, the integration of proprioceptive and vestibular information is vital for the accurate control of posture and balance [28]. Neck muscles also function synergistically with extraocular muscles during large gaze shifts involving eye and head movements [9]. It should be added that during the standing position, in which head movements do not occur, the proprioceptive and somatosensory information could play a greater role in maintaining postural control. The aim of this study was to evaluate whether the application of kinesio-tape on the SCM muscle could induce a perturbation of the postural control in healthy young adults, by altering the somatosensory/proprioceptive system of the neck.

The application of kinesio-tape on the SCM muscle belly seemed to involve some space-time parameters of postural balance, recorded by a computerised static force platform. The condition in which the kinesio-tape was applied bilaterally seems more to affect postural balance with respect to a unilateral application, both with and without visual afferences. Moreover, the highest effect on postural balance was observed in the open eyes condition compared to when the task was performed blindfolded. This seems to suggest that the disruption of the somatosensory system resulting from the application of the kinesio-tape was greater with the eyes open than with the eyes closed [29]. This may be because individuals were more dependent on vision than on proprioception to maintain their balance when their eyes were open making the proprioceptive system more susceptible to external perturbation in this condition. In fact, the proprioceptive information should not be considered as a system of isolated afferents but in relation to other systems such as vision and oculomotor afferents [30]. In contrast, in the absence of visual cues, there may have been a greater involvement of the muscle tendon receptors along with the entire proprioceptive system in maintaining postural control, which was only significantly disrupted when a higher perturbation (bilateral application of the kinesio-tape) affected the system [31]. Interestingly, this link between visual inferences and SCM strength was found during clenching tasks in subjects with myopia [32].

The CoP is the most used space-time parameter of the postural balance recorded by posturography, and it is indicative of the stability of the body mass with respect to the support base [33]. In our study, the length of the CoP parameter was significantly different in all experimental conditions compared to the control condition when the participants carried out the test with their eyes open. A different effect was observed when the participants performed the test with their eyes closed. In fact, the perturbation of the postural balance was significant compared to the control condition only when the kinesio-tape was applied bilaterally. In parallel, our results revealed that the maximal oscillation in the anteroposterior axis was statistically significantly longer as a result of the application of kinesio-tape. Additionally, with open eyes, the maximal oscillation was longer for the unilateral application of taping indicating a higher perturbation of the system when the stimulation was asymmetrical. Indeed, the entire CoP length increased with kinesio-tape, but there were no statistically significant differences between the three study conditions (bilateral application, left application and right application). These results suggest that an extrinsic stimulus over the SCM modified postural parameters similarly in all conditions tests.

Interestingly, despite the increase in the length of CoP and in the anteroposterior oscillation, the difference in the weight distribution between the left and right foot decreased when the participants performed the assessment with bilateral tape with respect to the control condition, whether with open or closed eyes. The application of the extrinsic stimulus (kinesio-tape) increased the oscillations but levelled the weight between the two lower limbs.

### Limitation

Our study has some limitations. We only investigated the immediate effect of kinesio-tape application to the SCM. The small sample size, the heterogeneity of demographic characteristics, and the presence of visual deficits do not allow generalizing the results. Therefore, future research should explore the long-term effects of kinesio-tape on postural balance with a large sample and randomized controlled trial design. Additionally, latent trigger points in the SCM muscle may lead to increased muscle fibre tension. Therefore, the possible presence of latent trigger points in the SCM of the study participants may have affected the individual postural balance. Moreover, multimodal instrumental evaluation of the vestibular system [34] and dynamic balance stability assessment can be performed.

## 5. Conclusions

Our results suggest that bilateral application of kinesio-tape on sternocleidomastoid muscles might be involved in postural control in healthy young adults. Bilateral tape application produced an increase in the oscillations, especially in the anteroposterior axis, but reduced the difference in the weight distribution between the right and left foot, with eyes open and closed. Moreover, the exteroceptive perturbation seemed to be greater in the somatosensory system when it was working coupled with visual afferences during upright position. These findings can be useful for further investigation into vestibular and neurological conditions characterized from postural control deficits.

## Figures and Tables

**Figure 1 healthcare-10-00946-f001:**
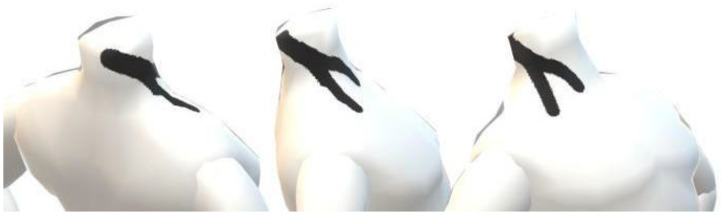
3D graphical representation of Kinesio-tape application (right side).

**Table 1 healthcare-10-00946-t001:** Demographic characteristics of the study sample. Abbreviation: SD: standard deviation; F: Female; M: Male; BMI: Body Mass Index; M: Myopia; A: Astigmatism. * Correction with glass or contact lenses.

Participant ID	Age (yrs)	Gender	Height (cm)	Weight (kg)	Foot Size (mm)	BMI	Refractive Deficit *
1	27	F	169	80	257	28.01	no
2	24	F	170	65	266	22.49	no
3	20	F	158	52	253	20.83	no
4	20	F	170	55	266	19.03	M/A
5	22	M	180	80	292	24.69	M
6	22	F	179	77	266	24.03	A
7	26	M	170	64	288	22.15	M/A
8	25	F	160	48	244	18.75	no
9	27	F	175	85	275	27.76	no
10	24	F	165	58	253	21.30	no
11	29	F	170	69	266	23.88	M/A
12	23	M	176	80	292	25.83	no
13	29	M	170	62	275	21.45	no
**Mean ± SD**	**24.46 ± 3.04**	**69.23 (F)**	**170 ± 7**	**67.31 ± 12.19**	**268.69 ± 15.29**	**23.09 ± 2.96**	**61.54% (No)**

**Table 2 healthcare-10-00946-t002:** Spatiotemporal parameters of static postural control. * *p*-value < 0.05.

Parameter	Left Tape	Right Tape	Bilateral Tape	Without Tape	Friedman Test	Post-Hoc	Post-Hoc	Post-Hoc
			**Open Eyes**					
	**L-SCM-KT**	**R-SCM-KT**	**B-SCM-KT**	**NO-KT**	** *p* ** **-Value**	**NO-KT vs.**	**NO-KT vs.**	**NO-KT vs.**
						**L-SCM-KT**	**R-SCM-KT**	**B-SCM-KT**
**Length of CoP**	1104.58 ± 361.2 *	1100.50 ± 392.5 *	1125.79 ± 351.6 *	846.86 ± 312.9 *	<0.0001	0.05	0.003	<0.0001
**Ellipse**	57.31 ± 46.1	130.19 ± 162	96.22 ± 148.1	101.86 ± 90.7	0.2	-	-	-
**X mm**	7.84 ± 38.3	−7.05 ± 10.5	−6.29 ± 8.4	−11.13 ± 8.28	0.06	-	-	-
**Y mm**	−10.50 ± 20.4	−12.17 ± 18.2	−7.73 ± 17.4	−9.73 ± 17.7	0.7	-	-	-
**Os. max (mm)**	3.25 ± 1.4 *	3.84 ± 3.9 *	2.97 ± 0.9 *	2.25 ± 0.6 *	0.004	0.01	0.03	0.01
**Os. min (mm)**	0.02	0.02	0.03	0.02	0.06	-	-	-
**RMS mm**	0.98 ± 0.3 *	0.99 ± 0.4 *	1 ± 0.3 *	0.75 ± 0.3 *	<0.0001	0.002	0.002	<0.0001
**RMS X mm**	0.54 ± 0.2	0.62 ± 0.4	0.60 ± 0.3 *	0.48 ± 0.2 *	0.03	-	-	0.02
**RMS Y mm**	0.80 ± 0.3 *	0.74 ± 0.2 *	0.77 ± 0.2 *	0.57 ± 0.2 *	<0.0001	0.001	0.007	<0.0001
**Weight L**	52.85 ± 4.4 *	53.08 ± 4.7 *	52.54 ± 4.6 *	55.08 ± 4.7 *	0.01	-	-	0.03
**Weight R**	47.15 ± 4.4 *	46.92 ± 4.7	47.46 ± 4.6 *	44.92 ± 4.7 *	0.006	0.04	-	0.01
		**Closed Eyes**						
	**L-SCM-KT**	**R-SCM-KT**	**B-SCM-KT**	**NO-KT**	** *p* ** **-Value**	**NO-KT vs** **.**	**NO-KT vs.**	**NO-KT vs.**
						**L-SCM-KT**	**R-SCM-KT**	**B-SCM-KT**
**Length of CoP**	1158.16 ± 370.9	1118.31 ± 383	1162.35 ± 392 *	923.92 ± 234.7 *	0.03	-	-	0.04
**Ellipse**	54.23 ± 60.39	160.73 ± 214.44	48.92 ± 40.2	50.90 ± 58.1	0.4	-	-	-
**X mm**	12.92 ± 62.14	−6.08 ± 9	−6.72 ± 8	−9.30 ± 6.6	0.1	-	-	-
**Y mm**	−8.12 ± 17.1	−10.50 ± 17	−8.34 ± 18.6	−9.26 ± 18.7	0.7	-	-	-
**Os. max (mm)**	3.57 ± 1.8	3.22 ± 1.3	3.36 ± 1.4	3.36 ± 2	0.5	-	-	-
**Os. min (mm)**	0.03	0.02	0.02	0.03	0.10	-	-	-
**RMS mm**	1.03 ± 0.3	0.99 ± 0.3	0.95 ± 0.4 *	0.81 ± 0.2 *	0.01	-	-	0.01
**RMS X mm**	0.61 ± 0.3	0.60 ± 0.3	0.57 ± 0.2	0.50 ± 0.2	0.8	-	-	-
**RMS Y mm**	0.80 ± 0.2	0.76 ± 0.2	1.82 ± 1.1 *	0.62 ± 0.2 *	<0.0001	-	-	<0.0001
**Weight L**	52.15 ± 4.4	51.61 ± 4.6	52.46 ± 4.5 *	54.31 ± 4.1 *	0.02	-	-	0.03
**Weight R**	47.85 ± 4.4	47.38 ± 4.6	47.54 ± 4.5 *	45.69 ± 4.1 *	0.02	-	-	0.04

**Table 3 healthcare-10-00946-t003:** Effect size.

Parameter	Open Eyes			Closed Eyes		
	NO-KT vs.	NO-KT vs.	NO-KT vs.	NO-KT vs.	NO-KT vs.	NO-KT vs.
	L-SCM-KT	R-SCM-KT	B-SCM-KT	L-SCM-KT	R-SCM-KT	B-SCM-KT
**Length of CoP**	0.76	0.71	0.83	0.75	0.61	0.73
**Os. max (mm)**	0.92	0.56	0.94	-	-	-
**RMS mm**	0.76	0.67	0.83	0.86	0.7	0.44
**RMS X mm**	0.3	0.44	0.47	-	-	-
**RMS Y mm**	0.9	0.85	1	0.9	0.7	1.51
**Weight L**	0.48	0.42	0.54	0.5	0.61	0.42
**Weight R**	0.48	0.42	0.54	0.5	0.61	0.42

## Data Availability

The data used to support the findings of this study are reported in this article.

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
