# Peer review of "The Effects of Sternocleidomastoid Muscle Taping on Postural Control in Healthy Young Adults: A Pilot Crossover Study"

_healthcare, 2022, doi:10.3390/healthcare10050946_

Round 1

Reviewer 1 Report

General comments This study aimed to determine whether applying Kinesio-tape to the SCM muscle can impair standing postural control by affecting the neck muscles' somatosensory system. This study would be great if the authors provided adequate references. For the materials, may authors explain that the materials have been calibrated or proven that materials in good quality Specific comments Line 40-48: For healthy individuals… of balance-related movement. Authors may put adequate references in each sentence. Line 50-53: Therefore… Authors may put adequate references in each sentence. Line 57-62: Part of this postural… to its previous position. Authors may put adequate references in each sentence. Line 68-69: Sternocleidomastoid muscle (SCM) is the predominant source of cervical VEMPs. Authors may put adequate references in each sentence. Line 73-75: In recent years, clinicians have often used kinesiology tape (Kinesio-tape) as a treatment strategy to support the neck fascia, muscles, and joints and reduce cervical pain. Authors may put adequate references in each sentence. Line 79-81: However, in most studies, the effects… Why do authors just put one reference instead plural references? Line 109-111: How does the study use good material (such as force monitoring) that has been calibrated? Any studies discussed the same materials? Line 171-176: This seems to suggest...in this condition (no references) Line 178-181: In contrast, in the…has affected the system (no references). Line 182-197: no references For the standing stability effect center of pressure (CoP), the authors may consider cit below article: # (2017). Effect of Postural Control Demands on Early Visual Evoked Potentials during a Subjective Visual Vertical Perception Task in Adolescents with Idiopathic Scoliosis. Front Hum Neurosci, 11, 326. # (2021) Variability of visually-induced center of pressure displacements is reduced while young adults perform unpredictable saccadic eye movements inside a moving room, Neuroscience Letters 764 (1) 136276

Author Response

General comments

This study aimed to determine whether applying Kinesio-tape to the SCM muscle can impair standing postural control by affecting the neck muscles' somatosensory system. This study would be great if the authors provided adequate references.

For the materials, may authors explain that the materials have been calibrated or proven that materials in good quality

Thanks for the revision, in the current manuscript we provided new references to support the background and the conclusion. Regarding the materials, we provide the commercial information about the force platform and kinesio-tape. We used the same force platform for other clinical trials and for the clinical assessment of our impatiens. The instrumentation is periodically (once a year) checked by specialized technicians. For the recording of spatio-temporal parameters we used a protocol, previously tested by other authors (Tamburella et al., 2014a; Tamburella et al., 2014b, Tamburella et al., 2013) and by our team (Lupo et al., 2018). We hope to find the new version of the manuscript in line with your suggestions.

Specific comments

Line 40-48: For healthy individuals… of balance-related movement.

Authors may put adequate references in each sentence.

Thanks, in the revised version of the introduction paragraph this sentence has been removed.

Line 50-53: Therefore… Authors may put adequate references in each sentence.

Thanks, we provided a reference.

Line 57-62: Part of this postural… to its previous position. Authors may

put adequate references in each sentence.

Thanks, we provided a reference.

Line 68-69: Sternocleidomastoid muscle (SCM) is the predominant source

of cervical VEMPs. Authors may put adequate references in each sentence.

Thanks, we provided a reference.

Line 73-75: In recent years, clinicians have often used kinesiology tape

(Kinesio-tape) as a treatment strategy to support the neck fascia,

muscles, and joints and reduce cervical pain. Authors may put adequate

references in each sentence.

Thanks, we provided two references.

Line 79-81: However, in most studies, the effects… Why do authors just

put one reference instead of plural references?

This sentence has been modified in the current version of the manuscript.

Line 109-111: How does the study use good material (such as force

monitoring) that has been calibrated? Any studies discussed the same

materials?

Thanks for this suggestion, all cited studies in the “Stabilometric Assessment” paragraph use the same instrument and the same procedure. We added further references about the force platform recording protocol.

Line 171-176: This seems to suggest...in this condition (no references)

Thanks, we provided a reference in support of our interpretation.

Line 178-181: In contrast, in the…has affected the system (no references)

Thanks, we provided a reference.

Line 182-197: no references

Thanks, we slightly modified the sentence and provided the reference “The CoP is the most used space-time parameter of the postural balance recorded by posturography and it is indicative of the stability of the body mass with respect to the support base. In our study, the length of CoP parameter was significantly different in all experimental conditions compared to the control condition if the participants carried out the test with their eyes open”.

For the standing stability effect center of pressure (CoP), the authors

may consider cit below article:

# (2017). Effect of Postural Control Demands on Early Visual Evoked

Potentials during a Subjective Visual Vertical Perception Task in

Adolescents with Idiopathic Scoliosis. Front Hum Neurosci, 11, 326.

# (2021) Variability of visually-induced center of pressure

displacements is reduced while young adults perform unpredictable

saccadic eye movements inside a moving room, Neuroscience Letters 764

(1) 136276

Thanks, we used this information in the new version of the manuscript. 

Reviewer 2 Report

The article titled : "Sternocleidomastoid Muscle Taping Involve Postural Control In Healthy Young Adults: a Four-condition Study" is interesting.

Remarks: The main weakness of this study is very low amount of participants only 13 who were divided into 4 subgroups. Therefore, I suggest add to the title - pilot study, because the conclusion made on 3 or 4 participants are not valid. In Limitation you should highlight it.

"13 participants (9F; 93 yrs 24.46±3.04) (other demographic information are available in Table 1) were randomly ex-posed, consecutively, at the four study conditions: (1) No kinesio-tape (Ctrl); (2) Ki nesio-tape on right SCM (R-SCM); (3) kinesio-tape on left SCM (L-SCM); (4) Ki-nesio-tape on both SCM (B-SCM)." This sentence should be change to made it more readable.

in the introduction: the sentence

"The aim of this study is to explore whether the application of kinesio-tape on the SCM muscle can induce a perturbation of the standing postural control in healthy young adults by altering the somatosensory system of the neck muscles". - I think that the word explore in this situation is too big.

Conclusions: 

"Our results suggest that bilateral application of kinesio-tape on sternocleidomastoid 214 muscles are involved in postural control in healthy young adults". - here also "are involved" is too big - might be involved.

Author Response

The article titled : "Sternocleidomastoid Muscle Taping Involve Postural Control In Healthy Young Adults: a Four-condition Study" is interesting.

Remarks: The main weakness of this study is very low amount of participants only 13 who were divided into 4 subgroups. Therefore, I suggest add to the title - pilot study, because the conclusion made on 3 or 4 participants are not valid. In Limitation you should highlight it.

Thanks for the important suggestion. We performed a cross-over study when each of 13 subjects randomly underwent all of the four conditions. We have clarified this point and added in the title “a pilot, cross-over” and explained it in the current abstract “Thirteen healthy participants (age: 24.46±3.04yrs; 9F) were enrolled and underwent in a randomly order to the four kinesio-tape (KT) conditions: without KT application (Ctrl); right KT application (R-SCM); left KT application (L-SCM); bilateral KT application (B-SCM).”.  

"13 participants (9F; 93 yrs 24.46±3.04) (other demographic information are available in Table 1) were randomly ex-posed, consecutively, at the four study conditions: (1) No kinesio-tape (Ctrl); (2) Ki nesio-tape on right SCM (R-SCM); (3) kinesio-tape on left SCM (L-SCM); (4) Ki-nesio-tape on both SCM (B-SCM)." This sentence should be change to made it more readable.

Thanks, we have changed the sentence: “Thirteen participants (9F; yrs 24.46±3.04) were tested in a random order to four conditions: (1) without kinesio-tape (Ctrl); (2) with Kinesio-tape on the right SCM muscle (R-SCM); (3) with the kinesio-tape on the left SCM muscle (L-SCM); (4) and with Kinesio-tape on both SCM muscles (B-SCM) (complete demographics of sample are available in Table 1).”

 in the introduction: the sentence

"The aim of this study is to explore whether the application of kinesio-tape on the SCM muscle can induce a perturbation of the standing postural control in healthy young adults by altering the somatosensory system of the neck muscles". - I think that the word explore in this situation is too big.

Thanks, we replaced "explore" with "test" in the current version.

Conclusions: 

"Our results suggest that bilateral application of kinesio-tape on sternocleidomastoid 214 muscles are involved in postural control in healthy young adults". - here also "are involved" is too big - might be involved.

Thanks, following your suggestion, we have changed the sentence: Our results suggest that bilateral application of kinesio-tape on sternocleidomastoid muscles might be involved in postural control in healthy young adults

Reviewer 3 Report

Evaluation of article number healthcare-1677595 with title " Sternocleidomastoid Muscle Taping Involve Postural Control In Healthy Young Adults: a Four-condition Study". The idea is interesting. There are editing errors and understatements in the article (see my comments). I have the following serious reservations:

  1. The group is not homogeneous in terms of sex and quite small? Why did the authors choose this group size?
  2. Why did the authors choose kinesio taping for healthy individuals? I don't see the practical application.
  3. Mixing people with refractive errors with healthy people is quite a big mistake. There is no information about the ophthalmologic examination whether the correction was properly chosen or whether they were people with myopia or hyperopia. Whether there were other visual disorders such as astigmatism. With such a small group, poor vision correction will significantly affect the results. Slightly sliding the glasses and looking non-centrally through the lens will also affect the results.

Specific comments

1 # L22 - "(age: 24.46±3.04yrs; 9k)" - "K" -if you are using the abbreviation for the first time, please expand it. 

2# L39-40 - “the act of maintaining, achieving, or restoring a state of balance during any posture or activity” - if this is a citation, please provide pages.

"For embedded citations in the text with pagination, use both parentheses and brackets to indicate the reference number and page numbers; for example [5] (p. 10). or [6] (pp. 101–105)."

https://www.mdpi.com/journal/healthcare/instructions

3# L41-42 - "Difficulties associated with maintaining standing balance manifest in case of pathology or injury, which can alter balance control" - Too much generality expand and add citations. What injuries and difficulties?

4# L45 - "various sensory receptors" - Once again. Too much generality please elaborate with proper citation.

5# L51-52 - "different sources of sensory orientation information" - I see that the authors are using large generalizations. Please write with precision. Such terms are not suitable for a scientific paper.  Please correct with proper citation.

6# L57-68 - What I'm missing here is information about disorders related to this system. Please elaborate.

7# L92-93 - "Participants without known pathologies that could interfere with the study were selected." - What do the authors mean when they write this statement? What were the pathologies? Who conducted the study?

8# L93 - "9F"- "K" -if you are using the abbreviation for the first time, please expand it.

9# Table 1. - Bold can only be on 1 line. In other lines, remove the bold. - https://www.mdpi.com/journal/healthcare/instructions

10# Table 1. - The following data are missing from the table: gender, what was the refractive error.

11# L100 - " A certified KT practitioner" -  Please provide full company name, location and country of manufacture.

12# L116-117 -" B-SCM). Each" - remove double space.

13# L118 - "1.5 meters away from them and three times with closed eyes (CE)." -  I'm glad the authors turned the aspect to changing the visual input. I suggest, therefore, that the introduction be expanded to include the possible effects of eye closure and opening on muscles, and the vestibulo-ocular reflex (VOR). I suggest you read the work published on mpdi:

DOI: 10.3390/jcm10225376

DOI: 10.3390/brainsci12010014

14# L122- . All participants with visual impairment performed the test wearing personal glasses or contact lenses" 133 - Because an uncorrected vision defect affects muscle tone. And this affects the balance. I repeat my request to specify how many people with the refractive error were and how this defect was. Next, please specify how the authors verified that the vision correction was appropriate? When was the last time the participants visited an ophthalmologist/optometrist?

15# L124 - " Statistical Analysis "  - Suggest adding effect size.

"Effect size helps readers understand the magnitude of differences found, whereas statistical significance examines whether the findings are likely to be due to chance. Both are essential for readers to understand the full impact of your work." - 10.4300/JGME-D-12-00156.1

16# L128 - "three repetition). All results with" -  remove double space.

17# L154 - Table 2. Results - In my opinion, this is not an appropriate title please change it.

18# Table 2. - Bold and underline can only appear on one line. Bold and underline must be removed from the other lines. Statistical significance, for example, mark "*". - https://www.mdpi.com/journal/healthcare/instructions

19# L182-184 - " The length of CoP is the most significant space-time parameter of the postural balance recorded by posturography and it is indicative of the stability of the body mass with respect to the support base" - For proof of words, please add quote.

20# L186 - "A different scenario" -  This is not a scientific statement please correct.

21# L204 - " Limitation"  - Suggests adding equal information about the heterogeneous and small study group.

22# References - References do not follow the style required in mpdi. All should be corrected.

"References should be described as follows, depending on the type of work:

Journal Articles:
1. Author 1, A.B.; Author 2, C.D. Title of the article. Abbreviated Journal Name Year, Volume, page range.

Books and Book Chapters:
2. Author 1, A.; Author 2, B. Book Title, 3rd ed.; Publisher: Publisher Location, Country, Year; pp. 154–196."

https://www.mdpi.com/journal/healthcare/instructions

23# In the work I observed self-citations with 15, 17, 19 numbers. I understand that authors are specialists in this field but in my opinion these self quotations can be avoided. Please remove them.

Thank you for the opportunity to review.

Author Response

Evaluation of article number healthcare-1677595 with title " Sternocleidomastoid Muscle Taping Involve Postural Control In Healthy Young Adults: a Four-condition Study". The idea is interesting. There are editing errors and understatements in the article (see my comments). I have the following serious reservations: 

Thank you for your revision, we are pleased that you found the idea interesting. 

  1. The group is not homogeneous in terms of sex and quite small? Why did the authors choose this group size?

Thanks for the interesting observation. We use convenience sampling for our study. We are aware of gender heterogeneity. Moreover, the number of participants involved in our study is in line with that involved in similar studies exploring the effects of kinesio-tape in healthy subjects (He et al., 2022; Hsiao et al., 2022; Yildiz et al., 2022; Naugle et al., 2021). We made some changes in the manuscript to reflect these amendments: “A convenience sampling of healthy individuals without known pathologies that could interfere with the study has been used”. About gender heterogeneity we included a sentence reminder in the limitation section: “The small sample size and the heterogeneity of demographic characteristics do not allow generalizing the results”.

  1. Why did the authors choose kinesio taping for healthy individuals? I don't see the practical application.

We carried out this study to preliminarily test the effects of kinesio taping on the sternocleidomastoid muscle of healthy subjects as, to date, no studies have explored this yet and available studies demonstrated that kinesio taping has effect on the contractile properties of muscles in healthy individuals (Yildiz et al. 2022). We also conducted this study to hypothesize what the effects of kinesio taping on the sternocleidomastoid muscles of people living with diseases may be. 

  1. Mixing people with refractive errors with healthy people is quite a big mistake. There is no information about the ophthalmologic examination whether the correction was properly chosen or whether they were people with myopia or hyperopia. Whether there were other visual disorders such as astigmatism. With such a small group, poor vision correction will significantly affect the results. Slightly sliding the glasses and looking non-centrally through the lens will also affect the results.

Thank you for raising this important point regarding the vision defects of the individuals involved in the studies. All subjects were affected by myopia and/or astigmatism adequately and appropriately treated by an ophthalmologist. Despite the presence of vision disturbances, the authors believe that the vision defects have not influenced the results of this study as the assessments were conducted with individuals wearing their glasses or contact lenses. In the current version, we added new information about refractive errors and discussed this limit.

Specific comments

1 # L22 - "(age: 24.46±3.04yrs; 9k)" - "K" -if you are using the abbreviation for the first time, please expand it. 

Thanks, we resolved it: “Thirteen healthy participants (age: 24.46±3.04yrs; 9 Female) [...]”.

 2# L39-40 - “the act of maintaining, achieving, or restoring a state of balance during any posture or activity” - if this is a citation, please provide pages.

Thanks, we added the page number. 

3# L41-42 - "Difficulties associated with maintaining standing balance manifest in case of pathology or injury, which can alter balance control" - Too much generality expand and add citations. What injuries and difficulties?

Thanks, we clarify this point with opportune citations: “Alterations of the balance control may be associated with ear symptoms, visual symptoms (i.e. diplopia and oscillopsia), vertigo and dizziness.  Vestibular disorders, chronic neurological diseases and musculoskeletal impairments might be frequently the causes of these symptoms. These symptoms should not be continuous but they might occur during head or body movements for exemple during activities of daily living and might become chronic.”

4# L45 - "various sensory receptors" - Once again. Too much generality please elaborate with proper citation.

Thanks, we clarify the three systems of receptors involved in postural control: “Visual, vestibular and somatosensory receptors”

5# L51-52 - "different sources of sensory orientation information" - I see that the authors are using large generalizations. Please write with precision. Such terms are not suitable for a scientific paper.  Please correct with proper citation.

We have rephrased the sentence with: “Therefore, postural control can be viewed as a closed-loop feedback control system with the integration of vestibular, visual and proprioceptive information for spatial orientation, which are regulated by the sensory integration process system”

6# L57-68 - What I'm missing here is information about disorders related to this system. Please elaborate.

Thanks for the observation. We reformule part of the background to clarify this link: “Therefore, postural control can be viewed as a closed-loop feedback control system with the integration of vestibular, visual and proprioceptive information for spatial orientation [10], which are regulated by the sensory integration process system [11]. The upright standing position is unstable as any external perturbation to the upright orientation pro-duces forces that accelerate the body. Deviations from the desired orientations are detected by the sensory systems (primarily somatosensory/proprioceptive, visual, and vestibular systems), which, in response, generate appropriate joint torques to preserve stability [11]. A deficit in one or more of these receptor systems or in the sensory integration can cause alterations of stability in upright postural control.”.

7# L92-93 - "Participants without known pathologies that could interfere with the study were selected." - What do the authors mean when they write this statement? What were the pathologies? Who conducted the study?

Thanks for this observation. We selected all participants that did not report neurological conditions or orthopedics conditions that impact the standing posture. We clarified it in the text: “[...] healthy individuals without history of neurological diseases or lower limb injuries that could interfere with the study has been included.”.

8# L93 - "9F"- "K" -if you are using the abbreviation for the first time, please expand it.

 Done

9# Table 1. - Bold can only be on 1 line. In other lines, remove the bold. -

Thanks, we resolved it.

 10# Table 1. - The following data are missing from the table: gender, what was the refractive error.

 Thanks, we added a column with gender information and more specifications for refractive error.

11# L100 - " A certified KT practitioner" -  Please provide full company name, location and country of manufacture.

Thanks, we added the information about the kinesio-tape manufacturer. “ [...] Kinesiology Human Tape, Prosomed®, Bolzano, Italy.” 

12# L116-117 -" B-SCM). Each" - remove double space.

 Done

13# L118 - "1.5 meters away from them and three times with closed eyes (CE)." -  I'm glad the authors turned the aspect to changing the visual input. I suggest, therefore, that the introduction be expanded to include the possible effects of eye closure and opening on muscles, and the vestibulo-ocular reflex (VOR). I suggest you read the work published on mpdi:

Thank you for your background suggestion, we use this information in the discussion section.

 14# L122- . All participants with visual impairment performed the test wearing personal glasses or contact lenses" 133 - Because an uncorrected vision defect affects muscle tone. And this affects the balance. I repeat my request to specify how many people with the refractive error were and how this defect was. Next, please specify how the authors verified that the vision correction was appropriate? When was the last time the participants visited an ophthalmologist/optometrist?

All participants involved in this study performed an oculist visit at least 24 month before the test. Each subject with visual deficits used optical corrections in line with the specialistic indication. Three out of the five subjects with visual deficits present a myopia and ametropia, one present myopia and the other ametropia; we provide this information in table I of revised manuscript. 

15# L124 - " Statistical Analysis "  - Suggest adding effect size. 

"Effect size helps readers understand the magnitude of differences found, whereas statistical significance examines whether the findings are likely to be due to chance. Both are essential for readers to understand the full impact of your work." - 10.4300/JGME-D-12-00156.1

Thanks for the interesting suggestion. We calculated the effect size using Cohen methods and we have integrated the new data in the methods, results and discussion paragraphs  of the current manuscript.

“The effect size has been calculated with Cohen method (Cohen’s d=Mean 1 – Mean 2 / √ (SD12+ SD22)/2) and interpret as small (d  =  0.2), medium (d  =  0.5), and large (d ≥ 0.8).”

16# L128 - "three repetition). All results with" -  remove double space.

 Done

17# L154 - Table 2. Results - In my opinion, this is not an appropriate title please change it.

Thanks, we renamed the Table 2. “Spatio-temporal parameters of static postural control”

18# Table 2. - Bold and underline can only appear on one line. Bold and underline must be removed from the other lines. Statistical significance, for example, mark "*". - 

Thanks, we resolved it.

19# L182-184 - " The length of CoP is the most significant space-time parameter of the postural balance recorded by posturography and it is indicative of the stability of the body mass with respect to the support base" - For proof of words, please add quote.

Thanks, we rewrite the sentence and provide a reference: “The CoP is the most used space-time parameter of the postural balance recorded by posturography and it is indicative of the stability of the body mass with respect to the support base [26]. In our study, the length of CoP parameter was significantly different in all experimental conditions compared to the control condition if the participants carried out the test with their eyes open”

20# L186 - "A different scenario" -  This is not a scientific statement please correct.

Thanks, we change “scenario” with “effect”: “A different effect was observed when the participants performed the test with their eyes closed”.

21# L204 - " Limitation"  - Suggests adding equal information about the heterogeneous and small study group.

Thanks for the comment, we added the information about  the heterogeneity and the limited sample size in the limitation paragraph: “The small sample size, the heterogeneity of demographic characteristics and the presence of visual deficits do not allow generalizing the results.” .

22# References - References do not follow the style required in mpdi. All should be corrected.

"References should be described as follows, depending on the type of work:

Journal Articles:

  1. Author 1, A.B.; Author 2, C.D. Title of the article. Abbreviated Journal Name Year, Volume, page range.

Books and Book Chapters:

  1. Author 1, A.; Author 2, B. Book Title, 3rd ed.; Publisher: Publisher Location, Country, Year; pp. 154–196."

https://www.mdpi.com/journal/healthcare/instructions

 Done

23# In the work I observed self-citations with 15, 17, 19 numbers. I understand that authors are specialists in this field but in my opinion these self quotations can be avoided. Please remove them.

Done

Thank you for the opportunity to review.

Round 2

Reviewer 1 Report

All the comments have been addressed in the newer verion of the manuscript.

Reviewer 3 Report

I congratulate the authors on their work in amending the article. The answers of the authors are acceptable. A small note please remove the comments on the right and bold the year in all of reference.

This manuscript is a resubmission of an earlier submission. The following is a list of the peer review reports and author responses from that submission.